# Interactive Camera Network Design Using a Virtual Reality Interface

**DOI:** 10.3390/s19051003

**Published:** 2019-02-27

**Authors:** Boris Bogaerts, Seppe Sels, Steve Vanlanduit, Rudi Penne

**Affiliations:** 1Faculty of Applied Engineering, University of Antwerp, Groenenborgerlaan 171, 2020 Antwerp, Belgium; seppe.sels@uantwerpen.be (S.S.); steve.vanlanduit@uantwerpen.be (S.V.); Rudi.penne@uantwerpen.be (R.P.); 2Department of Mathematics, University of Antwerp, Groenenborgerlaan 171, 2020 Antwerp, Belgium

**Keywords:** camera network design, virtual reality, submodular function maximization, camera placement, human factors

## Abstract

The traditional literature on camera network design focuses on constructing automated algorithms. These require problem-specific input from experts in order to produce their output. The nature of the required input is highly unintuitive, leading to an impractical workflow for human operators. In this work we focus on developing a virtual reality user interface allowing human operators to manually design camera networks in an intuitive manner. From real world practical examples we conclude that the camera networks designed using this interface are highly competitive with, or sometimes even superior to, those generated by automated algorithms, but the associated workflow is more intuitive and simple. The competitiveness of the human-generated camera networks is remarkable because the structure of the optimization problem is a well known combinatorial NP-hard problem. These results indicate that human operators can be used in challenging geometrical combinatorial optimization problems, given an intuitive visualization of the problem.

## 1. Introduction

In the camera placement problem, the goal is to find an optimal configuration of cameras to perform some observation task. Many practical problems can be formulated as an observation task, so it is no suprise that this problem has been studied extensively. For example, the design of surveillance systems for both large buildings and outdoor environments can be formulated as a camera network design problem [1,2]. In this formulation the objective is to maximize the camera coverage over a floorplan of the environment. The same problem arises in designing custom tracking systems in Virtual Reality applications. In the photogrammetry community a similar problem is studied, where the goal is to select image acquisition locations from which a 3D reconstruction will result in minimal uncertainty over reconstructed points [3,4,5]. In this formulation both coverage over environment points and expected measurement quality are important. The same formulation of the problem is also used in the field of inspection planning where reconstruction quality is generalized to arbitrary measurement acquisition functions [6,7,8]. In computer graphics the same problem occurs in view selection where the goal is to select a limited number of views (renders) of an object/scene that together provide the most efficient summary of information [9,10].

In general three distinct steps are important in automated camera network design: representation of the problem, formulation of the cost/quality function, and optimization of this cost/quality function. We will discuss these steps in Section 2 and highlight the fundamental problems in practical settings by studying its mathematical structure. We will present a virtual reality user interface where the user is in charge of placing all cameras, thereby avoiding all of the traditional steps, without loss of quality. We argue that this last finding is remarkable given the structure of the problem, which is also why we choose to cover this aspect in a separate section.

Our main idea, letting users design a camera network, is opposite to most approaches trying to automate this design. We will motivate that there are structural problems with automated design approaches that are not addressed properly. In this work, the strong spatial reasoning skills of humans, which is crucial in solving this design task, will be enhanced by the visualization possibilities of virtual reality (see Figure 1).

In Section 2.1 we will discuss the structural problems with automated camera design algorithms from both a mathematical and a user interactive perspective. In Section 4 will elaborate on our proposed interface that will be evaluated in Section 5.

## 2. The Automated Camera Network Design Problem

### 2.1. Problem Structure

As discussed earlier, the goal of camera network design is to find an optimal camera configuration that performs some observation task. In order to design a relevant camera network, knowledge about the environment is necessary, traditionally in the form of a CAD model which is frequently available. Further knowledge about the camera geometry and area of interest is strictly necessary. Early work on the art gallery problem tries to position cameras such that the entire area of interest is visible using geometric algorithms [1,11]. Approximate solutions are available for specific instances of this type of problem, that typically rely on a discretization of the problem [12].

Two things need to be discretized. Firstly, the area of interest should represented by a finite set of elements ie $E=\{e_{1},\ldots ,e_{n}\}$ (Environment). Secondly, the space of all possible camera locations needs to be a finite set of configurations ie $V=\{v_{1},\ldots ,v_{m}\}$ (Viewpoints). This discretization reduces the problem of finding a camera configuration that covers the area of interest, to the classical set covering problem (SCP) [13,14]. This binary representation of camera visibility is not really useful because it lacks the freedom to model realistic camera network performance models [6]. A more general problem formulation that encompasses the SCP is known as submodular function maximization [15]. This class of problems is well studied and a lot of complexity results of these problems are known.

### 2.2. Camera Network Design as Submodular Function Maximization

Submodular set functions are set functions that are characterized by a diminishing returns property. More formally if X,Y⊆Ω where Ω is a finite set, and X⊆Y. Let $f:2^{\Omega }⟶R$ a set function and x∈Ω∖Y be a singleton then submodular functions can be defined by following property:(1)f(X∪{x})−f(X)≥f(Y∪{x}−f(Y)

To relate the camera network performance models to submodular functions, we start by introducing a general formulation and proceed by giving concrete examples on how classical camera design modelling choices fit within this formulation. The formulation starts by defining a weight function $t:E\times N⟶R^{+}$ that for each environment point returns a quality value as a function of the number of cameras for which the point is visible. This visibility is formalized using a visibility function $f:E\times 2^{V}⟶N$ that computes for each camera configuration U⊂V and each environment point e∈E the number of cameras f(e,U) in *U* viewing *e*. The last function that we introduce fuses the quality function *t* with the visibility function *f* ie g(e,U)=t(e,f(e,U)). In this formulation the camera network design problem is formulated as maximizing the quality over all environment points with a limited set of cameras:(2)$$U^{*}=\underset{U\subset V:|U|=k}{\mathrm{argmax}}G\left(U\right)\;\mathrm{with}\;G\left(U\right)=\sum_{e\in E}g\left(e,U\right)$$

If t(e,−) is an increasing concave function, then *G* can be proven to be a monotone submodular function, thus the problem becomes the monotone submodular maximization problem with a cardinality constraint [15]. These technical requirements on *t* ensure that *g* exhibits a diminishing returns property, which translates in the intuitive notion that the importance of covering an environment point by an increasing number of cameras only decreases.

An example of such a *t* that incidentally transforms its associated submodular maximization problem to the SCP is:(3)$$t\left(e,n\right)=\left\{\begin{array}{cc} 0 & \mathrm{if}\;n=0\\ 1 & \mathrm{if}\;n>0 \end{array}\right.$$

Submodular function maximization is a combinatorial NP-hard problem that is well studied. An advantage is that there is an asymptotically optimal algorithm (we will use the more commonly used term ’optimal algorithm’ in the remainder of this work) to maximize a monotone submodular *G* under a cardinality constraint [16]. This algorithm is known as the greedy algorithm, and builds a solution by greedily adding the best cameras to the final solution set. The greedy algorithm yields a tight (1−1/e)-approximation (≈0.63-approximation) (here e is the base of the natural logarithm, not a problem specific constant), which means that the optimal function value is at most a constant factor (1/(1−1/e)) higher than the value returned by the greedy algorithm. The downside is that it is proven that no algorithm can improve on the greedy algorithm with a polynomial number of steps [16] which is exactly why it is called an optimal algorithm.

It is important to notice that the aforementioned results do not exclude superior optimizers that leverage some additional problem structure, but these algorithms will not generalize well and may be difficult to find. Furthermore, the geometric structure is hard to leverage as geometric algorithms have a prohibitive computational complexity related to the aspect graph [17]. These algorithms are typically not suited in general purpose camera network design applications.

### 2.3. Camera Network Performance Functions

An additional difficulty is that it is very challenging to formally define what exactly makes a camera system good. This notion of quality is also highly problem specific. An exact formulation is however needed in automated design approaches so a lot of propositions for such functions are available. In this section we will review popular function modelling choices and discuss their impact on the structure of the problem.

A first modelling choice is to encode the notion that it is better for an environment point to be viewed by multiple cameras [2,3,5,14]. This corresponds to extending the previously defined function *t* to arbitrary functions. The simplest function *t* is to simply count the number of cameras (cameras can have weights) and clip the function above some defined threshold [2], which results in a monotone submodular *G*. Another choice models the propagation of uncertainty in measuring environment points [3,5]. This formulation however does not guarantee a concave *t*, which makes the greedy algorithm less effective and makes the automated design problem provably more challenging. Note that *t* can depend on camera or viewpoint information.

Another modeling choice is to assign weights to viewpoints or/and environment points based on some notion of importance [2,10,14]. No weighting scheme can fundamentally change the problem structure as long as the weights remain positive.

Finally, regularizers can be added to the problem:(4)$$U^{*}=\underset{U\subset V:|U|=k}{\mathrm{argmax}}G\left(U\right)-\alpha \sum_{u_{k},u_{l}\in U}r\left(u_{k},u_{l}\right)$$

The positive constant α and positive regularizer function *r* are used as a tool to discourage the optimizer to choose certain undesired camera configurations [2]. An example where this can be useful is a stereo reconstruction camera system. Stereo reconstruction requires overlap in what neighbouring cameras perceive. The regularizer can penalize a lacking of overlap guiding the optimizer to solutions that exhibit this overlap. From the perspective of the problem structure this is however not a good idea. The regularizer completely destroys the monotonicity which will result in a far more challenging optimization problem with unbounded solution quality [15]. It is important to note that regularizers can pop up unexpectedly. Any element of a quality function that only depends on cameras and not environment points can be written as a regularizer (examples include: distance constrainsts, overlap constraints, orthogonality constraints, etc).

### 2.4. Solving the Automated Camera Network Design problem

A lot of strategies to solve the the Automated Camera Network Design problem are available in the literature. Next-best-view planning uses an optimizer that builds a solution by subsequently adding the best viewpoint to the solution set, which is exactly the greedy algorithm that is optimal for the problem [4,5,7,10]. Another approach is to employ an evolutionary optimization strategy [3]. A modular relaxation of the true submodular function is also used together with a branch-and-bound solution method [2]. The latter algorithms does not provide a theoretically better upper bound than the original greedy algorithm, so their performance is highly problem specific. Specialized branch-and-bound methods exist for submodular functions as well, but they suffer from the fact that submodular functions are much harder to bound than modular functions, which limits their practical applicability [15].

In this section we linked the camera network design problem to monotone submodular maximization with a cardinality constraint. We discussed manipulations of the problem as treated in camera network design literature and discussed their impact on the structure of the problem. This structure proves that automated algorithms are fundamentally limited in the quality of solution they can generate. An illustration why the asymptotic properties of algorithms are important is that the number of possible camera systems in the experiment in Section 5.1 is of the order $10^{115}$ (draw 25 cameras from 40,000 candidates). Because the greedy algorithm is optimal, any superior algorithm should address a significant subset of the huge number (as a reference, the estimated number of particles in the universe is only $3.28\times 10^{80}$) of possible solutions, which is not realistic.

## 3. User Interaction

As discussed earlier, the first step in most automated approaches is to discretize the inherently continuous camera network design problem. Discretizing the area of interest is in most cases straightforward. Strategies such as voxelization and random sampling of points exist and require very limited interaction from a user. Sampling of possible viewpoints is however much more difficult. Any viewpoint from this set can be chosen by the algorithm, so care should be taken that only valid configurations are present. This introduces following problems:The space of all possible camera positions must be represented by a finite number of samples that should be dense enough to accurately represent the problem.The number of samples should be low enough to avoid prohibitively long optimization times.Cameras cannot be placed at every position so a lot of domain specific knowledge is necessary to select these possible positions.

From an operational perspective this is not ideal because the user that should provide this information to the application needs domain specific knowledge, and at the same time, must know about sampling strategies which is rare. From a user interaction perspective there are also issues in finding a method that avoids having to select every point manually, but retains a qualitative sampling density.

While the user often has a good intuitive notion of what constitutes a good camera network, the choice of a quality function is challenging and highly problem specific. Furthermore, many methods have weights that need to be chosen that have a significant impact on the final result. Choosing this function with associated weights requires an intimate knowledge about the mathematics of a problem and the measurement specification of the cameras. Even then, iterations are necessary to align optimizer results with user expectations.

Defining a regularizer *r* and associated weight α is often required to avoid pathological behaviours of automated optimization algorithms [2] and to encode complicated network requirements. However, to our knowledge no literature exists on how to define these, while they will have a crucial impact on the final measurement system.

A general problem with all of the problems associated with user interaction and in our opinion the worst, is that all the information that needs to be provided is very abstract. This abstractness excludes practitioners from the adoption of automated algorithms for the design of camera systems.

User interaction for camera network design has not received a lot of attention in literature. A GUI has been proposed that gives the user tools to perform the required discretization and assign importance weights to environment parts [2]. In this tool, viewpoint sampling is limited to a uniform sampling over the region of interest which is not realistic in many real-world cases, and the cost function is fixed. A closely related work develops a GUI that allows users to manually choose camera configurations to complete 3D scans of single objects [18]. A user study was performed which showed that the quality of solutions provided by users and an automated algorithm are comparable, even with inexperienced users.

## 4. Virtual Reality Interface

### 4.1. Motivation and Overview

The basic principle of our virtual reality interface is simple. The user is placed in the scene of interest together with an initial camera setup *U*. The application will calculate g(e,U) for each environment point *e* and visualize these values as an interactive colored volume (cloud). Our choice of g(e,U) is discussed in Section 4.4 but it is important to note that this function is only needed to provide the user with a qualitative notion of quality. The user can manipulate all cameras and see the effect on the environment quality in real time. The user is in charge of generating a camera configuration, but can apply geometrical reasoning to solve this problem. Real-time color feedback of the values of g(e,U) allows the user to decide what is important for his application while performing the optimization.

A schematic overview of the difference between our proposed workflow and the traditional workflow is given in Figure 2. Our proposed workflow avoids the need to perform a challenging viewpoint discretization step and the step that designs a specific quality function. These steps both require intimate knowledge about both the practical problem and theory of camera design planning. The advantage of the traditional workflow is that a computer can automatically solve the camera network design problem. But in reality these results often do not correspond to what is expected by the user. This is mainly because the quality function is not aligned with the intuitive expectation of the user. Another common problem is that many informal notions of quality cannot be directly encoded in the cost function.

The structure of our interface (our implementation is publically available at: https://github.com/BorisBogaerts/V-REP-VR-Interface) consists of three main parts as shown in Figure 3. In this section, we will give a brief overview of their function but each will get a more detailed treatment in a later section. The first part is a simulator that handles all geometries, maintains its positions and performs dynamics calculations. The second part is responsible for calculating the coverage and quality for a given camera system. The final part is responsible for rendering to the virtual reality device and managing user interaction.

### 4.2. Simulator

As a simulator we use a commercially available robot simulator called V-REP [19], providing many features that are available to our interface. The use of a flexible simulator allows for the modelling of complicated real world dynamic systems up to a high fidelity and the subsequent design of camera systems in these environments.

In our proposed workflow, the simulator will also be responsible for the definition and modelling of the specific problem. Firstly, the geometry of the problem should be available in the shape of a triangular mesh. This data format is widely available by the use of CAD software packages in construction and design. When no CAD model is available, one could always resort to 3D reconstruction from image data. In our experiments section we will show both cases to highlight the flexibility of our solution. Next we require geometrical knowledge of each camera that can be positioned. This information consists of a perspective angle of the camera, the resolution, and the maximum/minimum measurement distance.

Finally, a discretization of the space of interest needs to be performed. We provide a box that can be positioned on the scene of which the size and position can be changed. This box is shown in purple, the left image of Figure 4. The user can select a discretization resolution which determines in how many voxels the box will be subdivided. From each voxel we select the center point and together these points determine the set *E*. We choose to represent *E* as points because geometric calculations using points are much faster than using objects that have volume.

### 4.3. Interactive Quality Computation

Interactively calculating quality values over the space of interest is a challenging task. The visibility between all environment points and all cameras in the scene need to be calculated in an environment of arbitrary complexity. To compute this visibility there are two options. The first is ray-tracing and the second is z-buffer visibility calculation. Because the first is slower with today’s hardware, we use the latter. This algorithm works by rendering the scene for a specific camera. For each rendered image there is an associated *z*-buffer which stores the depth of each pixel resulting from the rasterization algorithm used to render the image. Next, all points of interest are projected to the camera and get {u,v,z}-coordinates. These are pixel locations together with the depth value. Using these coordinates we can compare the depth value of each point with the depth value of the z-buffer at the same location. If the *z*-value of the point is smaller then the *z*-buffer value, the point is visible for the camera. In our results we were able to achieve computation times of the order of 300 ms to compute the quality of 50 k points of interest for 10 cameras in an environment of 300 k triangles. These results will be discussed more formally in the results section, but show the interactivity and scalability of the z-buffer approach. In our implementation we use the publically available implementation of this algorithm in the visualization-toolkit (VTK) [20].

The proposed method can also model cameras with a non-traditional field-of-view (e.g., omnidirectional cameras), as these field-of-views can be recreated by composing multiple perspective renders.

### 4.4. Virtual Reality Process

The final part of our application deals with managing the virtual reality component. This encompasses both the rendering of all information and managing the user interaction. In this process, it is important that the obtained frame rate of the rendering procedure is high enough to provide a comfortable virtual reality experience, so the focus of this part is speed. The information we render is:All problem geometryAn interactively rendered volumeVirtual camera feeds

As rendering engine we again use the visualization-toolkit (VTK) [20], which connects to the publicly available openvr-api which in turn connects to popular virtual reality hardware. The engine can handle real time volume rendering by using highly optimized implementation of GPU based ray-casting [21]. As a final feature the camera feeds obtained by rendering individual cameras as discussed in Section 4.3, are displayed as dynamic textures viewable by the user. These feeds contain useful information that the user can leverage during his design task. An example where this can be useful, is the design of live stereo reconstruction setups. In these setups it is important that adjacent cameras have enough overlap in what they perceive. In this application the user can see what each camera sees, and ensure himself that the required overlap is present.

The user is able to manipulate camera positions by dragging each camera with a controller. The virtual reality thread will update the position inside the simulator. This in turn results in a changing of the camera position in the volume thread. This eventually results in an updated quality function that changes the appearance of the volume. Furthermore, the user is able to change his position and scale relative to the scene. We believe that the latter is of fundamental importance to perform the design task. The user can for example do a rough initialization of the camera setup when the scene is small (zoomed out), and perform more detailed manipulations in a larger scene (zoomed in).

Finally, the user is able to choose a color/opacity function. The color/opacity transfer function selects a color and an opacity for every point in the volume as a function of its value. For example, the function gives an opacity of zero if the environment point is visible and red with opacity in the range 0<o<1 if a point is invisible. This example is exactly the color/opacity function of the right figure of Figure 1. In our implementation we support the choice of one user defined custom function that can be interactively changed. Our experience is that using two functions, one which shows the quality, and one that shows what is invisible, provides the user all information that is needed.

## 5. Experiments

To evaluate the performance of our proposed workflow we performed three experiments. The goal of the first two experiments is to evaluate the quality of the user-generated camera networks relative to automatically generated networks in different settings. In the third experiment we evaluate the computational performance of our interface to highlight its scalability. Images and basic information about each of the three experimental scenes is depicted in Figure 4. In all experiments we focus on realistic large scale and complex camera network design problems. The focus on more challenging design problems is important because simpler problems might give an overly optimistic view on human performance.

The performance of user-generated and computer-generated solutions are compared with the same quality functions and the same constraints in every experiment. To quantitatively compare different camera networks we use the same procedure to evaluate their quality *G*. We obtain this quality by computing a quality value *g* for each environment point *e* using the method discussed in Section 4.3. These local quality values *g* can finally be accumulated to a global quality value *G* by evaluating Equation (Equation 2). We do not focus on the time required to generate each solution because these times are relatively small compared to the total time needed to design and implement a camera system.

As automated algorithm we use the well known greedy algorithm [4,5,7,10] on a problem specific cost function. The viewpoint discretization of the problem is different in each experiment but is generally dense. This dense discretization ensures that the computer-generated solution is close to the best achievable solutions. In each experiment we actively searched for discretizations that resulted in the best performance for the automated algorithm.

### 5.1. Office Scenario

The goal of this experiment is to place 25 cameras in an office environment depicted in Figure 4 (middle). A top view of the entire scene is also shown in Figure 5. The objective of the camera system is to maximally cover the free space of the scene (quality function *t* is one if the environment point is covered by more than one camera, and zero otherwise). Cameras can be placed at ceiling level over the entire area of the office. The goal of this experiment is to evaluate the quality of user-generated solutions relative to computer-generated solutions in a setting where the cost function and constraints are clear and intuitive.

For this experiment we recruited 5 volunteers, that had no prior experience in designing camera networks. None of the users received any specific training prior to the experiment. All possible user interactions that are discussed in Section 4.4 were demonstrated just prior to the experiment. None of the users had difficulty with these interactions as these are sufficiently intuitive and simple. The users received the instruction to position the cameras as to maximally cover the free space. All uncovered areas were interactively displayed as a red cloud and visible for the users. Users were also instructed to position all cameras at ceiling level, which was evaluated during the experiments. If a camera was not placed at ceiling level they were instructed to re-position the camera. At the beginning of the experiment all cameras were placed outside the scene to eliminate the bias that any other initial setup might introduce. The users did not get a time limit and were instructed to stop when they were pleased with the camera configuration (no user spent more than 30 min designing the camera network (to give an indication about the timing)).

To obtain a set of viewpoints for the automated algorithm we uniformly and randomly sample 40 k points over the entire office at ceiling level. For each position we generate a random orientation that does not point upward. This discretization was the result of an active search for the best possible discretization achieving the highest coverage. All evaluations of coverage are relative to a dense sampling of the environment of 4.8 million samples (resolution of 0.1 m).

The computer-generated solution on the problem with initial discretization of 10 k samples achieved a coverage of 74%. Surprisingly, all users in the experiment performed better than the initial result of the automated algorithm. This is surprising in part because the challenging nature of the design problem and inexperience of the users. We also expected the scene to be too large (1600 $m^{2}$ with 25 cameras) for users to keep an overview of everything, which is necessary to find good solutions. After increasing the viewpoint sample size we were able to increase the performance of the automated algorithm to 77%, but users with our interface still scored comparable or better. Based on these experimental results we conclude that users with our virtual reality interface are at least highly competitive with automated algorithms and sometimes even better.

### 5.2. Harbour Scenario

In this section we will consider a real-world camera network design task where the camera network is used to guarantee safety. When containers are unloaded from ships, they are positioned on the harbour quay. After this unloading, workers have to confirm information on the container and thus have to move among these containers. To track these workers a camera system is attached to the crane to ensure that it can be safely operated. The scene used in our examples is depicted in Figure 4 (left).

The goal of this experiment is to evaluate the quality of user-generated solutions relative to computer-generated solutions in a more challenging yet more common setting. In this case the notion of quality is known by an expert and has to be encoded in a quality function. The quality function is also more complicated, making the design task more challenging for the user. This is also why we do not consider this problem solvable by non experts, instead we will focus on the performance of experts.

In this experiment we only had one expert at our disposal, who generated a solution and provided details about his notion of quality. The experimental conditions were the same as in the office scenario. The orange lines in Figure 4 indicate the possible positions of cameras in this problem which is limited to two beams on the crane. To construct a set of possible viewpoints, we linearly subdivide each beam in 20 positions and defined 15 possible orientations, creating a set of 2 × 20 × 15 (600) configurations. The goal is to select 10 positions that maximize coverage over the area of interest, defined as the purple box in Figure 4 and a denser sampling between the containers. This denser sampling encodes the notion that areas in between containers are more important because they are more dangerous. This box is discretized in 30 k points and between the containers we uniformly sample another 16 k points to force the optimizer to focus on areas between the container.

The design of a quality function is challenging because it has to encode qualitative user preferences. To reflect this point we will consider multiple quality functions. The expert preferred environment points to be covered by as many cameras as possible, but the added importance of a point being covered by more cameras is strictly decreasing. An example illustrating this notion of quality is that with an environment consisting of two points, both points being covered by two cameras is preferred over one point being covered by one camera and the other by three cameras. We can generate quality functions encoding this notion of quality by generating a sequence of 6 strictly decreasing values $\left\{s_{1},\ldots ,s_{6}\right\}:\sum_{i}s_{i}=1$. Each value $s_{i}$ is multiplied by the number of environment points that is viewed by more than i−1 cameras and summed together to create a quality function. Every function generated this way results in a monotone submodular *G*. In this experiment we will generate random quality functions $t_{i}$ and calculate for each function a camera network $U_{i}^{*}$. We also have a solution designed by an expert using our interface $U_{e}^{*}$.

The quality of different solutions $U_{i}^{*}$ versus $U_{e}^{*}$ will be evaluated by studying ratios $G_{i}\left(U_{j}^{*}\right)/G_{i}\left(U_{i}^{*}\right)$ and $G_{i}\left(U_{e}^{*}\right)/G_{i}\left(U_{i}^{*}\right)$. The former describes how well different quality functions agree on the quality of designed networks using a different quality function. This will give an indication of the loss in quality of solutions due to a misspecified quality function. The latter describes the quality of the user-generated solution with respect to different possible quality functions. A random subset of the obtained ratios is displayed in Figure 6. From these results we can conclude that the solutions of different quality functions tend to agree on other quality functions. There is a much larger difference between the quality of the user-generated camera network with respect to different quality functions. This means that for some quality functions the user-generated scores high, but for others it scores lower. On average automatically generated solutions are 11% better than the user-generated solution with respect to specific quality functions. If we compare the coverage (visible/total environment points) which is also important, the user-generated solutions score equal to or even better than automatically generated solutions. To compare the difference between an automatically generated network and a user-generated network, we show the obtained quality volumes for both solutions in Figure 7.

Figure 7 provides an insight why automatically generated solutions tend to score higher on specific quality functions. The automatically generated solution (right) focuses many cameras at areas of a larger volume. The user-generated solution focuses many cameras on small areas between containers, because these areas form a potential safety hazard which results in a lower overall function value. This indicates that scoring higher on a quality function does not necessarily results in better networks.

The expert noted that the quality of the solution he generated using our interface, is better than the automatically generated solutions despite the difference in quality as measured by the quality functions. This was due to following reasons:The user-generated solution featured more regular patterns of overlapStereo constraints between cameras are better in the user-generated solution which makes tracking algorithms more robust

Both remarks of the experts can be encoded in the quality function but can only be implemented as a regularizer (see Section 2.3). Based on these results we conclude that the user-generated solution is at least highly competitive with the automatically generated solution. This conclusion is based on the observation that if we are able to find the perfect regularizer, we can get closer to the expert’s requirements. This will inevitably result in a deteriorated performance of automated algorithms. In this case the quality gap between user-generated and computer-generated solution will typically only shrink.

### 5.3. Performance

It is important to note that we do not consider this as a formal performance test, there are too many variables that have an effect on the computational performance. The main goal is to convince the reader of the scalability of the interface. The experimental scenes introduced in Figure 4 have different areas and triangle counts and thus have a different performance. Other parameters that have an impact on the computation time are the number of cameras and the number of voxels. In this experiment we will focus on two parameters that indicate performance. The first which we call latency is defined as the time between a user moving a camera and the user seeing its effect on the volume. It is important to note that the latency does not affect the framerate in the virtual reality device because they operate in different computational threads. A second metric is the average framerate of the virtual reality device.

In this section we will evaluate both metrics on our implementation of the presented interface. Even though there is room for optimization of our code with respect to performance, the results will give a lower bound on the achievable performance. All experiments were performed on a computer with Nvidia GTX1070 GPU, intel I7-8700 CPU, and sufficient RAM (the application uses about 400 MB RAM). Both latency and average framerate are recorded during a walk through the scene. The latency is averaged over the entire run, but is independent of the position of the user.

In this experiment we add an additional scene which is the car park depicted in Figure 4 in the right image. This scene is reconstructed from images and features the highest triangle count with 530 k triangles. The images are obtained from the ETH3D 3D reconstruction benchmark [22] (the dataset consists of 44 images with resolution 6048 × 4032) and reconstructed using commercially available Autodesk ReCap Photo software.

The main results are summarized in Figure 8. These results are obtained by positioning 10 cameras of a resolution of 400 by 640 pixels in each scene. Average framerates for all scenes are over 40 FPS. We further noticed that the average framerate was independent of the number of voxels. The effect of the number of voxels on the latency is linear as expected, and remains linear up to a high number of voxels (2 million). We also increased the number of cameras up to 30 cameras and noticed the same linear trend. In the interest of saving space we therefore omitted this result from this work.

The framerates of the virtual reality device are enough to provide a pleasant virtual reality experience, even for scenes of up to 530 k triangles and a volume of 2 M voxels which qualifies as a large scale problem. The latency measured in our experiments indicates there is a clear linear trade-off between interactivity and accuracy (see Figure 8). The choice between both can be guided by the users preferences.

## 6. Discussion

The traditional recipe in the camera network design literature or related fields is to automate the design process, and to keep users away from this process. The motivation for this automation is that the task is too difficult for users to perform qualitatively. We however believe that this difficulty is not necessarily related to the problem structure. Automated algorithms also have structural difficulties with solving these problems as we have shown, and users can rely on strong geometrical reasoning which is not possible for computers. Using virtual reality we were able to visualize the quality of the camera coverage which is usually invisible and therefore augment users capabilities. The advantage of this approach is that users are much more involved with the task at hand, and the usual abstraction of automated algorithms is unnecessary. An additional advantage is that the user can translate what is important in a specific problem much more easily, which results in much fewer design iterations because he understands the problem.

We believe that the importance of our results transcend the camera network design literature. There are many problems in optimization that can uniquely be solved by experts. Often experts rely on an intuitive understanding of the problem to solve these issues. If this understanding is visual, a representation of this understanding in virtual reality can enable even non-experts to solve these problems. We have presented an example of such a visualization, for a challenging problem and shown that users are indeed capable of being competitive with automated approaches.

From our experiments we cannot conclude that users-generated solutions are systematically better than computer-generated solutions. We believe that if the problem gets more challenging, users can improve on the quality of automated algorithms. In further work we will study the submodular-orienteering problem, which is a combination of submodular maximization and the travelling salesman problem [23]. Automated algorithms are provably less efficient, but the problem is highly geometrical. This problem occurs in robot path planning for inspection tasks on which many communities rely (dimensional metrology, infrared thermography, etc.) [8].

## 7. Conclusions

We started by linking the camera network design problem to the monotone-submodular maximization problem with a cardinality constraint. Using this link we can conclude that the quality of solutions obtained by any automated algorithm is strictly bounded. Furthermore, for automated algorithms, to be able to solve the network design problem user interaction is required. This required user interaction is however very abstract, which is a big obstacle in the adoption of these algorithms.

In this work we proposed a virtual reality based user interface where the user can solve the camera network design problem manually, by applying geometrical reasoning. The workflow associated with this approach is much more intuitive and allows users without specialized knowledge to design camera networks. However, users with specialized knowledge can solve more specialized problems without knowledge about automated camera network design algorithms.

From our experiments we concluded that user specified camera networks are highly competitive with automatically generated solutions. We demonstrated this in two structurally different real-world camera network design problems. We also demonstrated the scalability of our approach to solve problems in geometries, resulting from 3D reconstructions from images up to high fidelity.

## Figures and Tables

**Figure 1 sensors-19-01003-f001:**
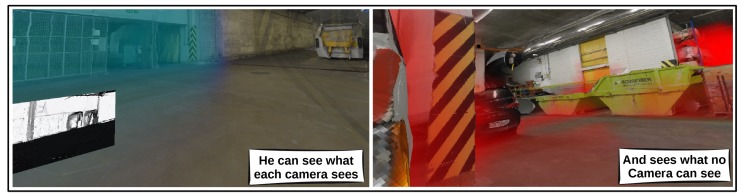
Screenshots from within our application showing that the user gets the supernatural ability to see what each camera can see (**left**), and regions that cannot be seen by the camera system (**right**), both as a colored volume. Using this information the user can change the positions of the cameras by dragging them and see the effect immediately. These capabilities allow users to design camera networks effectively.

**Figure 2 sensors-19-01003-f002:**
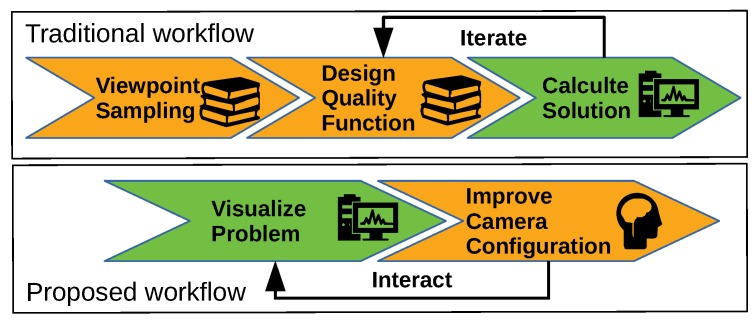
Orange arrows with a book pictograph indicate steps in a workflow that require specialist knowledge by the user. Orange arrows with a brain pictograph require visual reasoning of a user but no specialist knowledge, and the green arrows indicate work performed by a computer.

**Figure 3 sensors-19-01003-f003:**
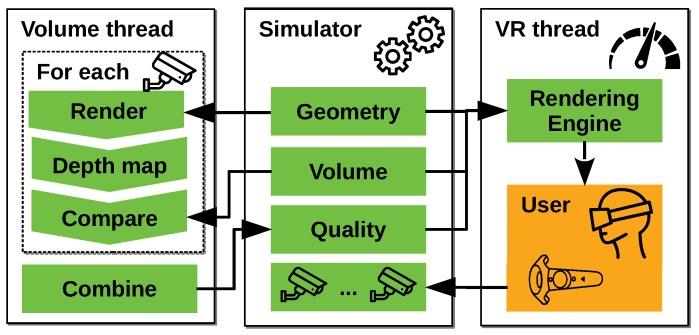
Three main parts that together form the structure of our interface.

**Figure 4 sensors-19-01003-f004:**
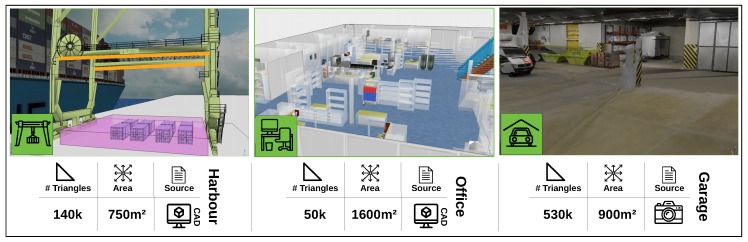
Overview of the various scenes used in our experiments and some associated metrics. The source of the scene is either from a CAD workflow or reconstructed from images.

**Figure 5 sensors-19-01003-f005:**
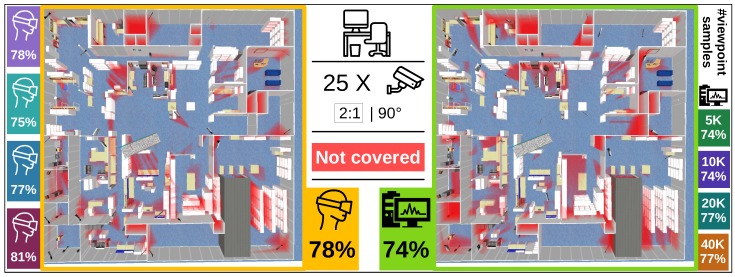
Top view of the office scene with uncovered areas shown in red. The left image displays a user-generated solution while the right shows a computer-generated solution. All percentages shown are coverage percentages, so higher is better. The camera network consists of 25 cameras with aspect ration 2:1 and perspective angle of 90 degrees.

**Figure 6 sensors-19-01003-f006:**
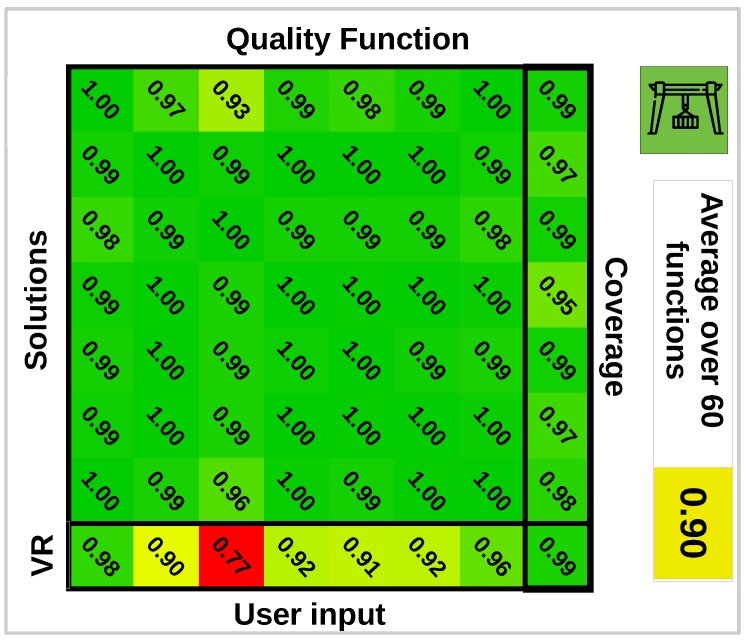
Table that cross evaluates solutions generated for different valid quality functions on these quality functions. The bottom row evaluates the user-generated solution on each quality function, and the right column shows the total coverage for each solution. This table does not represent every quality function we tested, but only a random subset. The total set contains 60 quality functions and their average is shown on the right.

**Figure 7 sensors-19-01003-f007:**
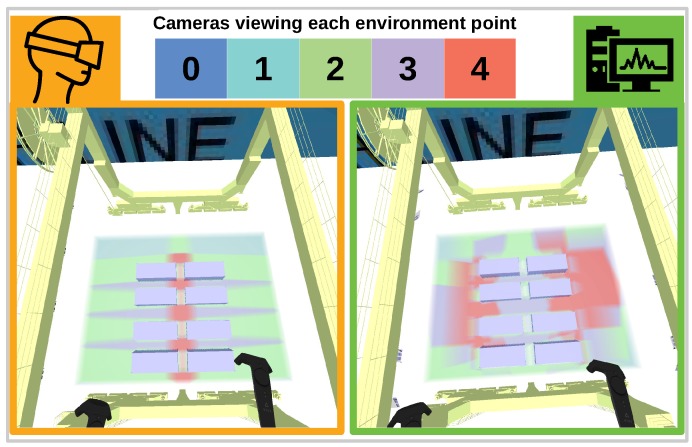
Comparison of a user-generated network (**left**) with a randomly chosen automatically generated network (**right**). The color values shown as a volume indicate for each point in the area of interest its redundancy (how many cameras see the point).

**Figure 8 sensors-19-01003-f008:**
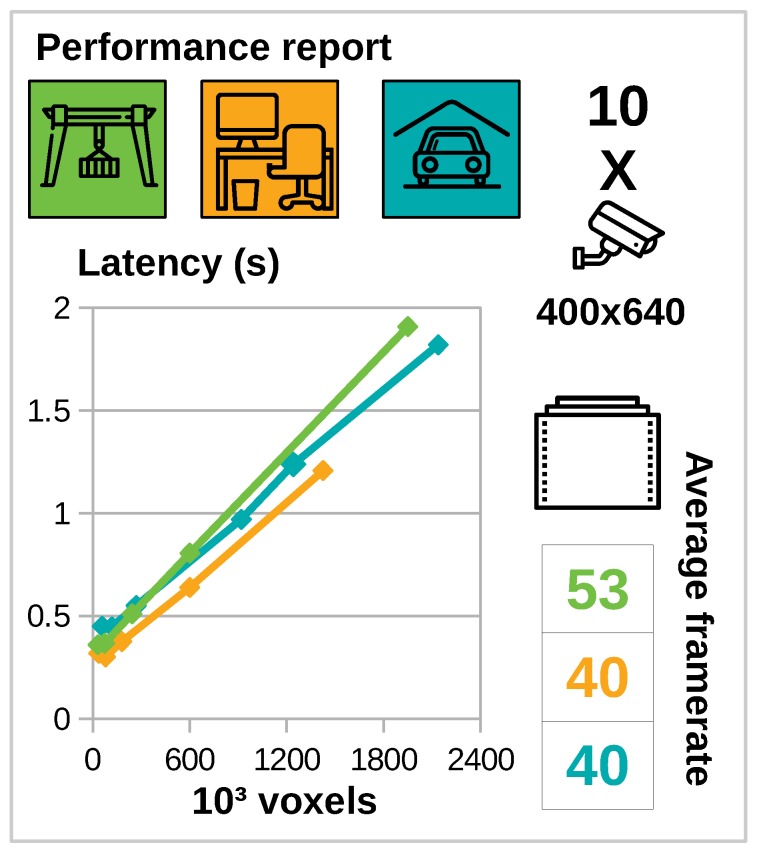
Summary of performance testing results. For different scenes we tested the effect of the number of voxels on the latency. We define latency as the maximum time between changing a camera position and a visible volume change. Latency times are for a scene with 10 cameras with a resolution of 400 by 640 pixels. Used symbols are introduced in Figure 4. Average framerate of the virtual reality thread are also included.

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
