# Peer review of "Interactive Camera Network Design Using a Virtual Reality Interface"

_sensors, 2019, doi:10.3390/s19051003_

Round 1

Reviewer 1 Report

This paper put forward a virtual reality user interface that allows human operators to manually design camera networks intuitively. Authors showed that the proposed human-in-the-loop design pipeline is competitive and sometimes superior than automated algorithms.

Experiments are thorough and convincing. Comparison between user-generated and algorithm-generated solutions have been conducted in multiple environments. 

The paper is well-written, language is good, figures are elegant. And the supplemental video and code release make the work even more complete. So I strongly supports the acceptance of this paper.

Questions:

1. In the experiments, it is mentioned that all volunteers have no previous experience with camera network design. I think it is useful to show what are the performance of expert camera network designers, as I imagine the system are developed for them in real applications.

2. It is also mentioned that there is no time limit for the human experiments. It will be informative to have these numbers.

Author Response

Open Review

English language and style

( ) Extensive editing of English language and style required
( ) Moderate English changes required
(x) English language and style are fine/minor spell check required
( ) I don't feel qualified to judge about the English language and style

Yes

Can be improved

Must be improved

Not applicable

Does the introduction provide sufficient background and include all   relevant references?

(x)

( )

( )

( )

Is the research design appropriate?

(x)

( )

( )

( )

Are the methods adequately described?

(x)

( )

( )

( )

Are the results clearly presented?

(x)

( )

( )

( )

Are the conclusions supported by the results?

(x)

( )

( )

( )

Comments and Suggestions for Authors

This paper put forward a virtual reality user interface that allows human operators to manually design camera networks intuitively. Authors showed that the proposed human-in-the-loop design pipeline is competitive and sometimes superior than automated algorithms.

Experiments are thorough and convincing. Comparison between user-generated and algorithm-generated solutions have been conducted in multiple environments. 

The paper is well-written, language is good, figures are elegant. And the supplemental video and code release make the work even more complete. So I strongly supports the acceptance of this paper.

Thank you for the positive comments.

Questions:

1. In the experiments, it is mentioned that all volunteers have no previous experience with camera network design. I think it is useful to show what are the performance of expert camera network designers, as I imagine the system are developed for them in real applications.

Response 1. We agree that expert camera network designers might perform better in the office scenario. We however did not have access to an expert for this experiment. We did however use an expert in the harbour scenario as mentioned on line 315.

2. It is also mentioned that there is no time limit for the human experiments. It will be informative to have these numbers.

Response 2. We explicitly avoided any discussion about design times to avoid confusion about the goal of our experiments, which is to compare the quality of the solutions. We do not focus on the design times because these, in reality, are dominated by encoding the many practical considerations in the abstract problem structure required for automated algorithms (pre-processing). A good comparison of design times would account for these times which are highly problem/expert dependent and thus are difficult to measure in an experiment.

We however agree that it is useful to have a rough indication of how long users needed to design the camera networks which is why we mentioned in footnote 7 that no user spent more than 30 minutes designing a camera network.

Reviewer 2 Report

A virtual reality user interface is developed in this work allowing human operators to manually design camera networks un an intuitive manner. Authors show that the developed method is competitive, and in some cases, it is superior to those generated by an automated algorithm. It as a well-written paper which shows an interesting work. However, some fundamental problems need to be addressed, for example, why human operator which is costly and time-consuming is considered? This makes us think that it may be unnecessary. Moreover, it is superior in some cases (not always). Therefore, how do you justify using human operation? Please consider my comments below to improve your manuscript.

It is not clear how the performance of these human-operated systems can be compared to the one previously published. For example, how the camera coverage is quantified. If you can quantify this, you need to briefly discuss the quantitative results in the abstract.

How the types of camera are affecting the results. For example, if the camera is an omnidirectional camera for some sections.

The automated methods should be quantified/discussed in the experiment/performance section. how do you prove that your method is better than automated methods (as discussed in the abstract)? If this comparison is not possible, please revised related section which you mentioned your method is superior to automated methods.

Automated methods can be more efficient in terms of cost and processing time (as described in lines 29-30). What is the main advantage of the human operator which can make this method superior to automated methods?  

Author Response

Open Review

English language and style

( ) Extensive editing of English language and style required
( ) Moderate English changes required
(x) English language and style are fine/minor spell check required
( ) I don't feel qualified to judge about the English language and style

Yes

Can be improved

Must be improved

Not applicable

Does the introduction provide sufficient background and include all   relevant references?

( )

(x)

( )

( )

Is the research design appropriate?

(x)

( )

( )

( )

Are the methods adequately described?

(x)

( )

( )

( )

Are the results clearly presented?

( )

(x)

( )

( )

Are the conclusions supported by the results?

(x)

( )

( )

( )

Comments and Suggestions for Authors

A virtual reality user interface is developed in this work allowing human operators to manually design camera networks un an intuitive manner. Authors show that the developed method is competitive, and in some cases, it is superior to those generated by an automated algorithm. It as a well-written paper which shows an interesting work. However, some fundamental problems need to be addressed, for example, why human operator which is costly and time-consuming is considered? This makes us think that it may be unnecessary. Moreover, it is superior in some cases (not always). Therefore, how do you justify using human operation? Please consider my comments below to improve your manuscript.

The replies to the reviewers remarks are covered in separate comments below.

It is not clear how the performance of these human-operated systems can be compared to the one previously published. For example, how the camera coverage is quantified. If you can quantify this, you need to briefly discuss the quantitative results in the abstract.

Response 1. In order to construct an automated algorithm it is necessary to define a function that quantifies how good a camera network is. An algorithm can than be constructed to generate a camera network that scores high on this function. Human-generated camera networks are also camera networks and can thus be used as input in the pre-defined function to receive a quantification.  In Sec. 4.3. we discuss how we evaluate compute the quality for each environment point. Eq. 2. explains that to obtain the total quality, we sum over all environment points. In this way we quantitatively compare results from user-generated and computer-generated (of any algorithm) camera networks. All our experimental conclusions are based on such quantitative comparisons (and thus also the ones mentioned in the abstract, see Fig. 5. and Fig. 6.).

Action: We explained how we compute the quality of camera networks, which allows us to quantitatively compare them in following added text in the experimental section:

To quantitatively compare different camera networks we use the same procedure to evaluate their quality $G$. We obtain this quality by computing a quality value $g$ for each environment point $e$ using the method discussed in Sec. 4.3., followed by evaluating Eq. 2

How the types of camera are affecting the results. For example, if the camera is an omnidirectional camera for some sections.

Response 2. We expect similar results for other types of cameras because the principle as to why humans are good at designing camera networks does not change. Humans have strong spatial/geometrical reasoning skills, so the only thing preventing them to design camera networks is that they do not understand the problem. Our interface can visualize exactly what the problem is, even if it involves omnidirectional, or other types of cameras. Our current implementation even supports this as it is possible to configure cameras in a cube-mapping like fashion (6 cameras, 90° fov, pointing in all principle axes) representing an omnidirectional field-of-view.

Action: We highlighted the technical capability of our proposed approach to deal with omnidirectional cameras by augmenting Sec. 4.3. with the following:

The proposed method can also model cameras with a non-traditional field-of-view (e.g. omnidirectional cameras), as these field-of-views can be recreated by composing multiple perspective renders.

The automated methods should be quantified/discussed in the experiment/performance section. how do you prove that your method is better than automated methods (as discussed in the abstract)? If this comparison is not possible, please revised related section which you mentioned your method is superior to automated methods.

Response 3. As automated method we used the greedy algorithm as discussed from line 266 and motivated in Sec. 2.4. In Sec. 2.  How we quantitively compare human-generated camera networks and computer-generated ones is discussed in Response 1.

Automated methods can be more efficient in terms of cost and processing time (as described in lines 29-30). What is the main advantage of the human operator which can make this method superior to automated methods? 

Response 4. SWe argue that a complete workflow to using automated methods does not only involve the processing time by the algorithm, but also a pre-processing step needed to tailor the abstract problem structure required by automated algorithm to the real world practical considerations. We highlighted several examples of this tailoring in the harbour case such as: highlighting more important regions, defining a good set of valid camera configurations, encoding requirements in a quality function, encoding inter camera constraints, etc.

Since all these extra steps are not necessary in our approach it will actually be faster in typical use cases. The omission of the extra steps also results in a workflow that is more accessible to non-experts. The decreased need for highly educated experts will also likely result in a cheaper workflow. We included a visual schematic representation of this argument in Fig. 2. We discussed the need for extra pre-processing and how it is solved traditionally Sec. 3.

Reviewer 3 Report

A well written paper: showing that human assisted VR can indeed better solve a practical and NP hard problem.

Review comments: A well written paper: showing that human assisted VR can indeed better solve a practical and NP hard problem.
The problem itself:
In the camera placement problem, the goal is to find an optimal configuration of cameras to perform some observation task. For example, the design of surveillance systems for both large buildings and outdoor environments can be formulated as a camera network design problem.
In this formulation both coverage over environment points and expected measurement quality are important. The same formulation of the problem is also used in the field of inspection planning where reconstruction quality is generalized to arbitrary measurement acquisition functions.
Advantages of this paper:
The competitiveness of the human-generated camera networks is remarkable because the structure of the optimization problem is a well known combinatorial NP-hard problem.
These results indicate that human operators can be used in challenging geometrical combinatorial optimization problems, given an intuitive visualization of the problem
Using this information the user can change the positions of the cameras by dragging them and see the effect immediately. These capabilities allow users to design camera networks effectively.
The authors present a virtual reality user interface where the user is in charge of placing all cameras, thereby avoiding all of the traditional steps, without loss of quality.
The authors will motivate that there are structural problems with automated design approaches that are not addressed properly. In this work, the strong spatial reasoning skills of humans, which is crucial in solving this design task, will be enhanced by the visualization possibilities of virtual reality.
While the user often has a good intuitive notion of what constitutes a good camera network, the choice of a quality function is challenging and highly problem specific.
The user can manipulate all cameras and see the effect on the environment quality in real time.
The proposed workflow avoids the need to perform a challenging viewpoint discretization step and the step that designs a specific quality function.
The advantage of the traditional workflow is that a computer can automatically solve the camera network design problem. But in reality these results often do not correspond to what is expected by the user. This is mainly because the quality function is not aligned with the intuitive expectation of the user.
As a simulator they use a commercially available robot simulator called V-REP.
In all experiments they focus on realistic large scale and complex camera network design problems.
Fig. 5 Evaluation result of one case study. Visibility coverage of 78% (human assisted) vs. 74% (automated).
For this experiment the authors recruited volunteers, that had no prior experience in designing camera networks. None of the users received any specific training prior to the experiment.
From their experiments they concluded that user specified camera networks are highly competitive with automatically generated solutions. The authors demonstrated this in two structurally different real-world camera network design problems. They also demonstrated the scalability of our approach to solve problems in geometries, resulting from 3D reconstructions from images up to high fidelity.

Author Response

Thank you for reviewing our manuscript.

question: However, the comparison is needed to show that your method outperforming automated methods, your latest modification improved the readability of your manuscript.

Answer: We compared our method with automated methods and discussed the result (lines 272-276, Sec. 5.1., Sec. 5.2., Fig.5., Fig.6.)

Round 2

Reviewer 2 Report

All questions have been answered, and some of the requested changes have been done. The evaluation is not quantified in the experiment by providing the comparison of the quality value g for each environment e as mentioned in lines 266-269. 

Author Response

Open Review

English language and style

( ) Extensive editing of English language and style required
( ) Moderate English changes required
(x) English language and style are fine/minor spell check required
( ) I don't feel qualified to judge about the English language and style

Yes

Can be improved

Must be improved

Not applicable

Does the introduction provide sufficient background and include all   relevant references?

( )

(x)

( )

( )

Is the research design appropriate?

(x)

( )

( )

( )

Are the methods adequately described?

(x)

( )

( )

( )

Are the results clearly presented?

( )

(x)

( )

( )

Are the conclusions supported by the results?

(x)

( )

( )

( )

Comments and Suggestions for Authors

All questions have been answered, and some of the requested changes have been done. The evaluation is not quantified in the experiment by providing the comparison of the quality value g for each environment e as mentioned in lines 266-269.

We agree that individual local qualities g do not allow for a quantitative comparison. In the experiments we used the global quality value G to quantitively compare camera networks. We hope to avoid this misconception by following modification (green part):

To quantitatively compare different camera networks we use the same procedure to evaluate their quality G. We obtain this quality by computing a quality value g for each environment point e using the method discussed in Sec. 4.3. These local quality values g can finally be accumulated to a global quality value G by evaluating Eq. 2.

Reviewer 3 Report

Can accept this paper. No more comments.

Round 3

Reviewer 2 Report

However, the comparison is needed to show that your method outperforming automated methods, your latest modification improved the readability of your manuscript.

Author Response

Open Review

English language and style

( ) Extensive editing of English language and style required
( ) Moderate English changes required
(x) English language and style are fine/minor spell check required
( ) I don't feel qualified to judge about the English language and style

Yes

Can be improved

Must be improved

Not applicable

Does the introduction provide sufficient background and include all   relevant references?

( )

(x)

( )

( )

Is the research design appropriate?

(x)

( )

( )

( )

Are the methods adequately described?

( )

(x)

( )

( )

Are the results clearly presented?

( )

(x)

( )

( )

Are the conclusions supported by the results?

(x)

( )

( )

( )

Comments and Suggestions for Authors

However, the comparison is needed to show that your method outperforming automated methods, your latest modification improved the readability of your manuscript.

We compared our method with automated methods and discussed the result (lines 272-276, Sec. 5.1., Sec. 5.2., Fig.5., Fig.6.)
